# Long-Term Effects of the Kumamoto Earthquake on Young Children’s Mental Health

**DOI:** 10.3390/healthcare11233036

**Published:** 2023-11-24

**Authors:** Masaharu Nagae, Eiko Nagano

**Affiliations:** 1Department of Clinical Nursing Sciences, Institute of Biomedical Sciences, Nagasaki University, Nagasaki 852-8520, Japan; 2Mifune Town Hall, Mifune-cho 861-3207, Japan

**Keywords:** children, disaster, earthquake, mental health, nursery school

## Abstract

Natural disasters cause numerous short- and long-term psychosocial effects on young children because of their increased vulnerability. This study aimed to examine the mental health of young children at 15 months after the Kumamoto earthquake. We conducted a self-administered questionnaire survey on the parents of 363 children aged 4–6 years across Kumamoto Prefecture. The questionnaire items included current residence, housing damage and evacuation experience during the disaster, as well as the Strengths and Difficulties Questionnaire (SDQ). The results showed that children who could stay in their home during the disaster had lower percentages of scores in the clinical range for conduct problems (odds ratio [OR] = 0.33, 95% confidence interval [CI]: 0.13–0.85) and hyperactivity/inattention (OR = 0.42, 95%CI: 0.19–0.93) on the SDQ. Furthermore, children who experienced living apart from their parents during the disaster had a higher percentage of scores in the clinical range for conduct problems (OR = 2.39, 95%CI: 1.05–5.42). At 15 months post-disaster, the mental health of the sample was worse than the normative data of Japan, indicating that the mental health of young children who experienced living at home and apart from their parents during the disaster was still affected.

## 1. Introduction

At 9:26 pm on 14 April 2016, a magnitude 6.5 earthquake struck Kumamoto, Japan. This was followed by a magnitude 7.3 earthquake at 1:25 am on April 16. Both these violent tremors occurred within a short period of time and triggered intense seismic activity, causing tremendous damage to the area. The disaster killed 228 people and injured another 2753. Furthermore, approximately 200,000 houses were destroyed. Additionally, more than 120 aftershocks with a magnitude of 4 or higher were recorded between April 14 and 30 [1]. However, a key characteristic of the Kumamoto disaster was that two major earthquakes occurred in the same area within a short period of time, which resulted in several aftershocks over a long period of time. Consequently, the health of the residents in the area was severely affected.

The effects of disasters are not only physical, but also social and psychological. In particular, psychological distress is common among disaster victims [2]. These psychological effects are more dramatic for children, as they are socially dependent and still developing both psychologically and physically [3]. Therefore, it is difficult for them to understand, judge, express and act in the face of various events. Children are the most vulnerable population to post-disaster mental health problems, as they have less experience and knowledge about how to cope with disasters and post-disaster events [3,4]. In children, the psychological effects of disasters may take the form of post-traumatic stress disorder (PTSD), depression, anxiety, emotional distress, sleep disturbance, increased fear, internalising and externalising behaviours, separation anxiety, event re-enactment and increased emotional numbness and arousal [5]. Influencing factors include the child’s physical proximity to the disaster, the extent to which the disaster affected their home, family and the environment, their socio-economic status, the presence of disability, minority and social status, disruption of family functioning due to the disaster in the community as a whole, parents’ psychopathological symptoms and post-earthquake stress in parents [6,7,8,9,10,11]. Apart from the impact of the disaster itself, the effects of secondary post-disaster stressors, such as separation from family and pets, inadequate food and water supplies, destruction of housing, relocation and interrupted school attendance, have also been noted [12,13].

PTSD and depression are often used as indicators of the psychological impact of a disaster on children; however, caution should be exercised while using these indicators for children under the age of 6 years because it is difficult for children of that age to express their psychological and physical symptoms verbally [14]. It has also been indicated that parents and caregivers may not fully understand internal distress in children, and that if the adults being assessed are themselves experiencing stressors from the disaster, their impeded capacity may affect their assessment of the children’s symptoms [2,15]. Besides PTSD and depression, studies of infants post-disaster have identified several other symptoms, such as clinginess, dependency, sleep disturbance (e.g., nightmares, reluctance to sleep alone, crying at night), irritability, aggressive behaviours, incontinence, hyperactivity, separation anxiety, apathy, playful re-enactment, anniversary reactions, fear of sounds and places, regression phenomena and changes in appetite and defecation patterns [8,16,17]. Apart from assessing children’s psychological responses through their behavioural changes, some studies have examined their overall mental health in terms of behavioural characteristics using the Strengths and Difficulties Questionnaire (SDQ) [18,19].

Although some studies have shown that PTSD and clinical symptoms decline over time, many studies on children under the age of 14 years have shown that these symptoms, in fact, do not decrease over time, or once reduced, may undergo re-aggravation [2]. Moreover, it is presumed that children affected by disasters show different psychological responses at different developmental stages, and that the younger the age, the greater the impact of parental psychological symptoms and the secondary stressors associated with reduced parenting time [16]. However, very few disaster studies have been conducted with preschool-aged children (under the age of 6 years) as compared to studies with school-going children and adolescents [2,4,8]. Therefore, to assess and support the mental health of young children after a disaster, the extent to which preschool children’s mental health recovers over time needs to be clarified. Moreover, it is crucial to identify the secondary stressors that strongly affect the safety and security of young children, such as separation from parents and changes in the living environment. Given this background, the present study aimed to clarify the family/housing and mental health statuses of preschool children at 15 months after the Kumamoto earthquake. Additionally, to identify the children who may need long-term support, we examined the long-term effects of the stressors experienced by the children during the Kumamoto earthquake or its evacuation procedures.

## 2. Materials and Methods

### 2.1. Study Design

In Town A in Kumamoto Prefecture, a screening survey of children in all nursery schools and certified childcare centres in town was conducted. Town A, located about 3 km from the epicentre of the Kumamoto earthquake, was one of the most affected districts. First, a survey of all nursery schools in Town A was conducted in June 2016 (2 months after the disaster) to assess the situation of households with children and plan individual support [20]. Based on the survey results, infants and their parents in Town A were provided support in collaboration with nursery schools and experts, and a follow-up survey was conducted at 1 year after the first survey in July 2017 (15 months after the disaster). In the follow-up survey, the SDQ was added to the survey items to assess children’s mental health more objectively. This study analysed the mental health of young children and the long-term effects of the earthquake at 15 months after the disaster using the follow-up survey data.

### 2.2. Participants

The parents of 756 children aged 0–6 years enrolled across six nursery schools and two accredited childcare centres in Town A of Kumamoto Prefecture were invited to participate in the follow-up survey, among whom 710 accepted. We analysed the data of 363 children between the ages of 4 and 6, the target age range for the SDQ. We excluded the data of 37 children due to incomplete responses and finally analysed the data for 326 children.

### 2.3. Procedure

The survey consisted of a self-administered questionnaire, which was conducted on the participating parents of Town A. The data were collected in July 2017, 15 months after the earthquake. The Town A staff explained the survey to the heads of each facility and requested co-operation. Each facility then explained the survey to the parents and requested co-operation. The survey forms were distributed to the parents of the children from each facility, completed by the parents and then collected by the facilities. Finally, the town office collected all the forms and recorded the data. The data were then anonymised and analysed by the researcher.

### 2.4. Instruments

#### 2.4.1. Demographics

Demographic factors included children’s age and sex; living conditions included current residence (own house, own rented house, grandparent’s or relative’s home, temporary housing and temporarily rented public housing) and family type (nuclear or extended); social support included the people that parents could consult (spouse, parents, parents of spouse, siblings, relatives, friends, nursery or schoolteachers, medical professionals, neighbours, no support and others).

#### 2.4.2. Damage Situation and Evacuation Status during the Disaster

The respondents were asked about the damage to their home as a result of the disaster (not destroyed, partially destroyed, half-destroyed, majority destroyed, completely destroyed and others) and their children’s experience of evacuation during the disaster (living in their home, staying in a car, living in an evacuation shelter and living apart from their parents). In the ‘others’ category of the house damage section, seven respondents were identified as being a part of a ‘long-term evacuee household’, which was defined as ‘a household with a housing unit that has become uninhabitable because of the continuing danger of damage due to pyroclastic flow, etc., caused by the natural disaster or for other reasons, with the uninhabitable state being likely to last for a long period’ [21]. For the purpose of public assistance, these households were treated in the same way as those that were ‘totally destroyed’.

#### 2.4.3. Strengths and Difficulties Questionnaire (SDQ)

To assess children’s adjustment and mental health status objectively and comprehensively, the Japanese version of the SDQ (parent-completed, for children aged 4–17 years) was used for children aged 4 years and older [22]. The SDQ contains four subscales related to difficulties (conduct problems, hyperactivity/inattention, emotional symptoms and peer relationship problems) and one subscale related to strengths (prosocial behaviour), with each subscale consisting of five items, amounting to a total of twenty-five items. Each item is graded on a three-point scale as follows: 0 points for ‘not true’, 1 point for ‘somewhat true’ and 2 points for ‘certainly true’. The score range for each subscale is 0–10 points. A higher score for the difficulty subscale indicates greater difficulty, and a higher score for the strength subscale suggests greater strength. Matsuishi et al. [23] confirmed the validity and reliability of parental assessments for 4–12 year olds and found that the total difficulty score (TDS) ranged from 0 to 40. In the present study, the TDS and five subscales were the main outcomes. The scores were categorised into normal, borderline and clinical ranges (0–12, 13–15 and 16 points or more for TDS; 0–3, 4 and 5 points or more for conduct problems; 0–5, 6 and 7 points or more for hyperactivity/inattention; 0–3, 4 and 5 points or more for emotional symptoms; 0–3, 4 and 5 points or more for peer relationship problems; and 6–10, 5 and 4 points or fewer for prosocial behaviour, respectively) with the clinical range used as a cut-off.

### 2.5. Statistical Analysis

First, simple tabulations were created for each item. The distribution of each score on the SDQ is shown for the normal, borderline and clinical range groups. Three classifications are commonly used on the SDQ to obtain an overview of the mental state; however, a borderline range group may not necessarily mean that support is required. Therefore, to clarify the influencing factors for children in the clinical range who may be eligible for support, the two groups were further divided by clinical range. Six outcomes (clinical range of each score on the SDQ) were examined. Multivariate logistic regression was used to calculate the adjusted odds ratios (ORs) and 95% confidence intervals (95% CIs). The variables considered in the models were age, sex, current residence, damage to home due to the disaster and evacuation experience at the time of the earthquake. SPSS (v29; SPSS Inc., Chicago, IL, USA) was used for the statistical analysis, with the level of statistical significance set at 5%. For model validation, the Hosmer–Lemeshow goodness-of-fit test was used, with which it was identified that the model was well adjusted if the *p*-value was greater than 0.05, and the closer to 1, the better the fit.

### 2.6. Ethical Considerations

The purpose of the survey was declared on the survey form. The explanation of the survey and the request for co-operation were given by the Town A staff to the heads of each facility and from them to the children’s parents. Consent to publish the results of the analysis consisting of anonymised data was obtained through an opt-out method. The publication of the results of this study was approved by the Ethics Committee of the Nagasaki University Graduate School of Biomedical Sciences (approval No.: 18061430).

## 3. Results

### 3.1. Sample Characteristics

The participants’ (*n* = 326) demographic characteristics, damage situation and evacuation experience at the time of the earthquake are summarised in Table 1. There were 162 (49.7%) female and 164 (50.3%) male children; 131 (40.2%) were age 4 years, 161 (49.4%) were age 5 years and 34 (10.4%) were age 6 years. As for their current residence, 256 (78.5%) children lived in their own home and 70 (21.5%) were living in places other than their own home; 248 (76.1%) lived in nuclear families and 78 (23.9%) lived in extended families. Regarding housing damage, 225 (69.0%) of the houses were less than or equal to half-destroyed, whereas 90 (27.6%) were more than half-destroyed. Eleven respondents (3.4%) selected the ‘others’ category in terms of the damage to their home at the time of the disaster. Expanding on their choice, seven of these respondents wrote ‘We had lived in long-term evacuation households’; two respondents cited they ‘Built a house after the earthquake’; one wrote ‘Not certified by the government as damage’; and one did not give any explanation. Eleven of the ‘other’ cases were excluded from further analysis because housing damage could not be determined from their responses. Moreover, 53 (16.3%) children had experienced living apart from their parents. The people that the parent could consult were their spouses and parents, followed by friends, siblings and parents of the spouse. Only 10 (3.1%) of the respondents could consult medical professionals, and 2 (0.6%) said they had no one to consult with.

### 3.2. Relationship between SDQ Scores and Each Factor

A summary of the SDQ scores is shown in Table 2. In terms of the TDS for overall difficulty, the mean (standard deviation [SD]) TDS was 8.53 (4.01) for female and 10.06 (4.57) for male children. Moreover, 9 female (5.7%) and 24 male (15.2%) children were in the clinical range (16 points or more).

Each score on the SDQ was divided into two groups by clinical range. The results of the comparisons for each factor are shown in Table 3. Consequently, we found a statistically significant difference between the two groups. The group of males had higher percentages of scores in the clinical range for the TDS (OR = 3.23, 95%CI: 1.41–7.37), conduct problems (OR = 2.12, 95%CI: 1.01–4.43), hyperactivity or inattention (OR = 2.34, 95%CI: 1.18–4.63) and prosocial behaviour (OR = 1.81, 95%CI: 1.01–3.25). The group of children that could stay in their home during the disaster showed a lower percentage of clinical range scores in conduct problems (OR = 0.33 [95%CI, 0.13–0.85]) and hyperactivity or inattention (OR = 0.42 [95%CI, 0.19–0.93]) on the SDQ. Furthermore, the group of children that experienced living apart from their parents during the disaster had a higher percentage of clinical range scores in conduct problems in the SDQ (OR = 2.39 [95%CI, 1.05–5.42]). However, no statistically significant differences were found based on current residence status or damage to home due to the disaster. The results of the Hosmer–Lemeshow test showed a *p*-value of 0.302 for the TDS, 0.591 for conduct problems, 0.705 for hyperactivity/inattention, 0.154 for emotional symptoms, 0.959 for peer relationship problems and 0.176 for prosocial behaviour, indicating that each model had an acceptable goodness-of-fit.

## 4. Discussion

Currently, studies on the long-term effects of earthquake damage on children’s mental health are limited. Therefore, to clarify the long-term effects of the Kumamoto earthquake, this study examined the relationship between the mental health of young children at 15 months after the earthquake and the situation at the time of the disaster.

### 4.1. Family and Mental Health Statuses of Preschool Children at 15 Months after the Kumamoto Earthquake

In this study, only 34% of the children were able to live in their own home during the disaster; however, 15 months later, 78.5% of the children were able to live in their own home. This indicated that the recovery from the disaster had progressed steadily. By contrast, about 10% of the children were still living in temporary housing or temporarily rented public housing even 1 year after the earthquake, indicating that it could take more than 1 year for the complete reconstruction of damaged houses. Putting special processes and subsidies in place to fast-track the reconstruction of severely affected houses could alleviate this long-term stressor.

To assess the mental health of young children aged 4 years and older at 1 year after the disaster, the SDQ was used. The mean (standard deviation [SD]) TDS was 8.53 (4.01) for female and 10.06 (4.57) for male children, with 33 children (10.5%) being in the clinical range. In the Japanese normative data, the mean (SD) TDS for children aged 4–5 years was 6.98 (6.0) for females and 7.94 (7.0) for males, and the clinical range for children aged 4–12 was 9.5% [23,24]. In comparison with these normative data, it can be said that the participants in this study tended to have poorer mental health. This suggests that, even over 1 year after the earthquake, the children’s mental health was still in the process of recovering. In particular, 9 female (5.7%) and 24 male (15.2%) children in the clinical range needed to be assessed by specialists; therefore, the need for support should be re-examined.

### 4.2. Long-Term Effects of the Stressors Experienced by Children during the Kumamoto Earthquake or Subsequent Evacuation Procedures

The present findings suggest that the experiences of not being able to live at home during a disaster and of living apart from their parents may affect children’s conduct and hyperactivity or inattention problems, even 1 year after a disaster. Home provides a foundation for a child’s life and is a symbol of safety and security. In the immediate aftermath of a disaster, most people are helpless and frightened by such a sudden and unprecedented situation, and living in a residence that is not their own home affects the psychological and physical functioning of not only children, but also their parents [20]. When facing this, parents can alleviate children’s fears in a protective way and help them regulate their behaviours and emotions in the face of stressors [11,25]. Particularly, the recovery of children aged 4–6 years who are experiencing stress is facilitated by obtaining a ‘sense of security and safety’ from their parents, their emotional safety base. Previous studies have shown that separation from family members due to a disaster has a significant impact on children’s psychological well-being after a disaster [8,12]. The results of this study suggest that these experiences may have an impact on the children’s mental health not only immediately after the disaster, but also 1 year later. Because children are socially dependent and still developing psychologically and physically, they encounter difficulties in understanding, judging, expressing and acting in the face of various events. Therefore, it is necessary to consider the possibility that the traumatic experiences of an earthquake and tsunami, even more than 1 year later, may cause problems in conduct, such as hyperactivity or inattention that are often expressed by a child. It is therefore crucial to examine the necessity of careful observation and professional treatment.

By contrast, no differences were found in terms of current residence, damage to the home at the time of the disaster or living in shelters. Previous studies have shown that the destruction of houses and extent of housing damage due to disasters have long-term effects on children’s psychological well-being [7,8,12,14]. However, these effects were not observed in the present study. Town A in Kumamoto Prefecture began recovery activities immediately after the disaster. Major roads were restored, emergency temporary housing was built, damaged houses were repaired and public disaster housing was built for those who had lost their homes. Furthermore, the government provided subsidies for the costs of repairing and rebuilding houses if they were certified as being half-destroyed or more. Even if a house is severely damaged by a disaster, the psychological impact of the event may be reduced by offering compensation to those affected for securing or rebuilding their home as soon as possible.

Research on the long-term effects of disasters is insufficient, and in particular, there are no studies focusing on infants and toddlers. Wang [2] noted that the first year of the disaster, particularly the first 6 months, is the peak time for symptoms and problems to arise. Without timely and effective interventions, a disaster’s impact on children’s development across the region can be long-lasting and severe. Psychosocial interventions in the immediate aftermath of a disaster are critical, especially for young children. Parents can help alleviate their children’s fears in protective ways and assist them in regulating their behaviours and emotions in the face of stress [11,25]. When parents themselves are unable to cope with their own symptoms adequately following a disaster, they may lack the necessary material and emotional resources to help their children [26]. In the immediate aftermath of a disaster, parents are also exposed to a crisis situation and are likely to suffer from mental health problems, often taking it out on their children [20]. Besides providing immediate physical and psychological support in the aftermath of a disaster, continuous monitoring and support are required to address the psychological damage inflicted on young children and their parents.

### 4.3. Limitations

This study has several limitations. First, the SDQ is an assessment tool that is based on parental perceptions and is considered to be reliable because it is administered to parents who are closely involved in their children’s lives. However, it is not a direct assessment of children’s mental health, so caution is needed when interpreting the results because parents’ mental health symptoms may affect the assessment of their children. Second, we could not clarify how many of the 710 children were living in the same household. If the same parent was reporting on multiple children, this would have confounding effects on the results. Third, the survey items were not sufficiently detailed because the purpose of the survey was not to test hypotheses. Although the survey assessed the magnitude of housing damage caused by the earthquake and whether the respondents had experienced evacuation, we did not obtain information on damage other than that to houses, the duration of evacuation over the course of the 15 months or the content of support received, which limits the interpretation of the results. Specifically, the lack of variables on the duration of circumstances and the inability to consider interaction effects between the level of damage to one’s home and the experience of living in one’s home during the disaster is a major limitation of this study.

In the immediate aftermath of a disaster, it is difficult to conduct surveys and research, as the top priority is to provide support to the victims. However, in the future, longitudinal surveys should include items for the kind of support people may need; such a study may lead to a better understanding of the recovery of children and their parents after a disaster. Although we did not have access to sufficient data for the present study, even if it is not a hypothesis-testing type of research, cumulative analysis of such valuable post-disaster data could reveal the long-term effects of disasters on children’s mental health.

## 5. Conclusions

In the present study, we evaluated the mental health status of preschool-aged children at 15 months after the Kumamoto earthquake and examined the long-term effects of stressors that they experienced during evacuation. As a result, at 15 months post-disaster, the mental health of the sample was worse than the normative data of Japan, indicating that the mental health of young children who experienced living at home and apart from their parents during the disaster was still affected.

## Figures and Tables

**Table 1 healthcare-11-03036-t001:** Children’s (*n* = 326) demographic characteristics, damage situation and evacuation experience during the disaster.

		*n*	%
Sex	Female	162	49.7
	Male	164	50.3
Age (years)	4	131	40.2
	5	161	49.4
	6	34	10.4
Current residence	Own house—owned	169	51.8
	Own house—rented	87	26.7
	Grandparents’ or relatives’ home	29	8.9
	Temporary housing	25	7.7
	Temporarily rented public housing	16	4.9
Family type	Nuclear family	248	76.1
	Extended family	78	23.9
Damage to home due to the disaster	Not destroyed	73	22.4
	Partially destroyed	152	46.6
	Half-destroyed	61	18.7
	Majority destroyed	12	3.7
	Completely destroyed	17	5.2
	Others	11	3.4
Experience of living in their home during the disaster	Yes	111	34.0
	No	215	66.0
Experience of staying in a car during the disaster	Yes	226	69.3
	No	100	30.7
Experience of living in an evacuation shelter during the disaster	Yes	87	26.7
	No	239	73.3
Experience of living apart from their parents during the disaster	Yes	53	16.3
	No	273	83.7
People with whom a parent can consult (Multiple answers, from the parent’s perspective)	Parents	263	80.7
	Spouse	258	79.1
	Friends	212	65
	Siblings	145	44.5
	Parents of spouse	103	31.6
	Teachers	44	13.5
	Relatives	32	9.8
	Neighbours	28	8.6
	Medical professionals	10	3.1
	Others	15	4.6
	No support	2	0.6

**Table 2 healthcare-11-03036-t002:** Overview of scores on the Strengths and Difficulties Questionnaire (SDQ).

		All (*n* = 315)		Female (*n* = 157)		Male (*n* = 158)	
		N	%	*n*	%	*n*	%
Total difficulty score (TDS)	Normal range	243	77.1	129	82.2	114	72.2
	Borderline range	39	12.4	19	12.1	20	12.6
	Clinical range	33	10.5	9	5.7	24	15.2
Conduct problems	Normal range	236	74.9	124	79.0	112	70.9
	Borderline range	41	13.0	19	12.1	22	13.9
	Clinical range	38	12.1	14	8.9	24	15.2
Hyperactivity/inattention	Normal range	223	70.8	115	73.2	108	68.4
	Borderline range	45	14.3	26	16.6	19	12.0
	Clinical range	47	14.9	16	10.2	31	19.6
Emotional symptoms	Normal range	271	86.0	136	86.6	135	85.4
	Borderline range	21	6.7	10	6.4	11	7.0
	Clinical range	23	7.3	11	7.0	12	7.6
Peer relationship problems	Normal range	279	88.6	147	93.6	132	83.5
	Borderline range	25	7.9	7	4.5	18	11.4
	Clinical range	11	3.5	3	1.9	8	5.1
Prosocial behaviour	Normal range	191	60.6	96	61.1	95	60.1
	Borderline range	61	19.4	35	22.3	26	16.5
	Clinical range	63	20.0	26	16.6	37	23.4

**Table 3 healthcare-11-03036-t003:** Odds ratios (ORs) and 95% confidence intervals (CIs) for damage situation and evacuation status during the disaster associated with clinical range scores on the SDQ ^1^ (*n* = 315).

	*n*	Total Difficulty Score			Conduct Problems			Hyperactivity/Inattention		
		OR	95%CI	*p*	OR	95%CI	*p*	OR	95%CI	*p*
Sex										
Female	157	1	(reference)							
Male	158	3.23	1.41–7.37	0.005	2.12	1.01–4.43	0.047	2.34	1.18–4.63	0.015
Age (years)										
4	127	1	(reference)							
5	155	0.58	0.27–1.29	0.182	0.54	0.26–1.15	0.111	0.51	0.25–1.05	0.066
6	33	0.53	0.14–2.03	0.359	0.40	0.11–1.53	0.18	1.47	0.57–3.77	0.430
Current residence										
Their home	251	1	(reference)							
Other than home	64	1.20	0.45–3.22	0.715	0.82	0.32–2.09	0.673	1.51	0.65–3.55	0.339
Damage to home due to the disaster										
Less than half-destroyed	225	1	(reference)							
More than half-destroyed	90	0.89	0.36–2.21	0.803	1.33	0.58–3.02	0.501	0.7	0.31–1.59	0.393
Experience of living in their home during the disaster										
No	208	1	(reference)							
Yes	107	0.72	0.30–1.69	0.445	0.33	0.13–0.85	0.022	0.42	0.19–0.93	0.033
Experience of staying in a car during the disaster										
No	97	1	(reference)							
Yes	218	1.21	0.52–2.79	0.66	1.24	0.57–2.74	0.587	1.69	0.79–3.64	0.177
Experience of living in an evacuation shelter during the disaster										
No	233	1	(reference)							
Yes	82	1.45	0.65–3.24	0.369	1.23	0.57–2.64	0.605	1.44	0.72–2.91	0.304
Experience of living apart from their parents during the disaster										
No	263	1	(reference)							
Yes	52	1.28	0.48–3.42	0.622	2.39	1.05–5.42	0.038	1.69	0.76–3.76	0.203
Hosmer–Lemeshow test				0.302			0.591			0.705
	** *n* **	**Emotional Symptoms**			**Peer Relationship** **Problems**			**Prosocial Behaviour**		
		**OR**	**95%CI**	** *p* **	**OR**	**95%CI**	** *p* **	**OR**	**95%CI**	** *p* **
Sex										
Female	157	1	(reference)							
Male	158	0.98	0.41–2.34	0.965	3.24	0.8–13.05	0.099	1.81	1.01–3.25	0.046
Age (years)										
4	127	1	(reference)							
5	155	1.05	0.42–2.62	0.926	0.63	0.18–2.19	0.463	0.67	0.37–1.21	0.183
6	33	0.86	0.17–4.3	0.855	0	-	0.998	0.4	0.13–1.28	0.122
Current residence										
Their home	251	1	(reference)							
Other than home	64	0.76	0.2–2.96	0.692	1.68	0.29–9.71	0.563	0.93	0.41–2.11	0.855
Damage to home due to the disaster										
Less than half-destroyed	225	1	(reference)							
More than half-destroyed	90	0.85	0.28–2.58	0.767	0.19	0.02–1.83	0.15	0.52	0.24–1.12	0.094
Experience of living in their home during the disaster										
No	208	1	(reference)							
Yes	107	1.88	0.78–4.55	0.161	1.24	0.33–4.67	0.753	0.57	0.3–1.1	0.093
Experience of staying in a car during the disaster										
No	97	1	(reference)							
Yes	218	2.05	0.67–6.31	0.21	0.48	0.14–1.67	0.248	0.79	0.43–1.46	0.455
Experience of living in an evacuation shelter during the disaster										
No	233	1	(reference)							
Yes	82	0.7	0.22–2.16	0.528	0.64	0.12–3.36	0.599	0.77	0.39–1.5	0.441
Experience of living apart from their parents during the disaster										
No	263	1	(reference)							
Yes	52	0.54	0.12–2.42	0.419	0.69	0.08–6	0.738	1.38	0.66–2.92	0.393
Hosmer–Lemeshow test				0.154			0.959			0.176

^1^ SDQ, Strengths and Difficulties Questionnaire.

## Data Availability

The datasets analysed during the present study are not publicly available because the data from this survey belong to Town A.

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
