# Peer review of "Long-Term Effects of the Kumamoto Earthquake on Young Children’s Mental Health"

_healthcare, 2023, doi:10.3390/healthcare11233036_

Round 1

Reviewer 1 Report

Comments and Suggestions for Authors

Thank you for the opportunity to review the submission of the journal article “Long-term effects of the Kumamoto earthquake on young children’s mental health” (#2606935) in the International Journal of Environmental Research and Public Health.

This article poses relevant questions regarding a population that is under-studied, it is well aligned with the journal’s scope and could be published after relatively minor revisions. I made a number of recommendations to improve the article. I divide my reviews into key points and minor comments, as most of the minor comments are issues that can be fixed relatively quickly, but major points often need further thought.

Key Points:

1.       Method: Table 2. I would recommend displaying the variables in Table 2 in the same format as that used for the analysis in Table 3. For example if categories were used as binary variables in Table 3 it may be more appropriate to also group them in Table 2.

2.       Method: It would be good to correct for multiple comparisons, at the moment it is not clear how many regression models were used (6?), and whether a correction (e.g., Bonferroni) is appropriate.

3.       Discussion: The focus at the start of the discussion should be on summarising the key study findings. Perhaps it would be best to move some of the content pertaining to the number of social supports experienced by parents further down in the discussion – as this was not a variable used in the prediction model.

4.       Overall: The study is an important contribution to the field; the descriptive responses alone would be worth publishing. I wonder if adding a few tables as an appendix that can better capture the summary statistics for the main study variables would be a worthwhile pursuit?

Minor Comments:

1.       Introduction, line 35: (…) in the area “was” severely affected (…)

2.       Introduction, line 38: Please provide a citation for the statement regarding more dramatic psychological effects in children.

3.       Introduction, line 43: Specify that the psychological effects “may” take the form of …

4.       Introduction, lines 74-76: Provide supporting evidence for the statement regarding greater impacts experienced by younger children.

5.       Method, line 109: conducted “by” the participating parents “on” Town A.

6.       Method, line 124: It would be good to know whether “medical professionals” would have included psychologists, social workers, etc. or whether it would have strictly been limited to doctors.

7.       Method, line 130: It is unclear whether there could have been other options, such as living with a relative, living in temporary accommodation with their parents, etc.

8.       Method, table 1: Perhaps consider renaming the variable Parents (within the “people to consult question” at the bottom of table 1) into Grandparents. Ideally all labels would be either from the perspective of the child or from the perspective of the parents.

Comments on the Quality of English Language

The English was good, there were a few sentences that needed correction which were not pointed out by comments above, but these were slight mistakes.

Reviewer 2 Report

Comments and Suggestions for Authors

Thank you for doing a study on an important topic.

How did you decide on 15 months? Was that timeframe arbitrarily chosen? Was it based on previous research as a good time to do follow up? Some explanation as to how it was chosen would be a good addition to your paper.

In the body of your paper, you identify your town as "Town A"; however, in your acknowledgments, you thank Mifune Town Hall which I believe reveals the name of Town A. 

In Tables 3-1 and 3-2, you are not consistent in using the lower-case "p" in the headings. Some of them are written as upper-case "P". 

Although I am not qualified to evaluate statistical analyses as I am not an expert in it, I think many of your p-values are quite high. Can you comment?

Comments on the Quality of English Language

In line 35, please add "were" between "area" and "severely" ("...in the area were severely affected.")

Reviewer 3 Report

Comments and Suggestions for Authors

I found the aims of this paper to be important and well-justified by the literature. I think the authors provide meaningful contributions to the literature regarding child adjustment following natural disasters. However, I think moderate revisions are warranted to strengthen the paper. 

ABSTRACT

The presentation of the odds ratios within text are confusing; it is unclear which group each result is associated with, and who the reference groups are. Clarification is needed. 

INTRO

line 32-34 is redundant with information presented imediatly prior in the introduction. 

line 34-35 is missing a word (in the area was severely affected?)

line 50 the authors note "psychopathological issues" - this should be reworded to say symptoms in order to be less stigmatizing. 

line 62-67 needs supporting citations

METHOD

It appears that the parents of 756 children were invited to participate and 710 participated. I would reframe the current wording, since there were not 756 participants (unless I am mis-interpreting the paragraph, in which case it should be reframed for clarity)

The authors should clarify how many of the 710 children were in the same household, or that these were 710 individual households

Relatedly, in the analyses, nesting of children within households was not taken into account. If the same parent is reporting on multiple children, this will have confounding effects on results. This was not mentioned in limitations either, but is an important consideration which either needs to be addressed in the analyses or noted as a limitation in the discussion.

line 137 authors note 7 children were excluded due to not being able to determine housing damage; is this 326-7 or were the 7 excluded before reaching the total of 326? If they were included after, which analyses were they excluded from, because the tabulations include them. If they were removed before, the participants section should note that 7 were excluded for this reason. 

The authors refer to sex and gender interchangeably throughout the paper; they should be clear about which term they assessed and are referring to, and use the correct one consistently instead of conflating the terms. 

Line 165 notes that regression was used "after controlling simultaneously for potential confounders" but fails to identify what the confounding variables are vs the variables of interest. The subsequent sentence lists all included variables without differentiating which is a control and which is a variable of interest. 

RESULTS

The sample included a disproportionately small percentage of 6-year-olds but the authors do not address why this is, or how this may affect results.

The authors repeatedly note that the SDQ was divided into two groups based on clinical range, but table 2 includes borderline range as well; the authors should clarify if borderline was folded into normal range. Additionally, they should consider the value of including borderline range in Table 2, since it is not discussed anywhere in the paper. 

The regression results included significant differences based on age and gender; these should be noted in the results and discussion. 

How were changes in circumstances over the course of the 15 months taken into account in the regression analyses? For example, a child may have been living in a car for 12 months and then currently living at home, and this would likely impact them significantly more than a child who was living in their car for one month and then returned to home. Same would be true for duration of time away from their parents; the authors should at least note that lack of variables on duration of circumstance is a strong limitation of the findings and an important confounding variable. 

Similarly, there is likely an interaction between the level of damage to one's home and their experience of living in their home during the disaster. A child who has more than half their home destroyed but lives in it will likely be impacted in very different ways than a child who has more than half destroyed but does not have to reside in the home. The authors should consider the interaction effect between these two variables. 

In table 3 it would be helpful for the authors to highlight significant findings in some way (e.g. bold, asterisk next to p value, etc). 

DISCUSSION

The authors emphasize the importance of social support in the discussion, yet barely include this variable in the analyses. It was also not included as any part of the aim of the study. It is unclear why this variable is included in the present study. The authors should either integrate the variable into the regression analyses as a covariate or remove it from the paper. 

lines 333 "as a result..." and 334 "moreover" are nearly identical sentences; needs editing to remove redundancy. 

Round 2

Reviewer 1 Report

Comments and Suggestions for Authors

Thank you for your revisions

Reviewer 2 Report

Comments and Suggestions for Authors

Looks good in present form. Thank you for the revisions.

Reviewer 3 Report

Comments and Suggestions for Authors

I appreciate the authors' efforts to revise the manuscript following my initial review. I found their revisions to sufficiently address concerns, and believe it has significantly increased the quality of the paper. 

Author Response

We appreciate your insightful comments. We worked hard to be responsive to you. Thank you for taking the time and energy to help us improve the paper.